# Vitamin C and E antioxidant supplementation may significantly reduce pain symptoms in endometriosis: A systematic review and meta-analysis of randomized controlled trials

Patrick Bayu ⓘ *, Jacobus Jeno Wibisono

Department of Obstetrics and Gynecology, Faculty of Medicine, Pelita Harapan University, Tangerang, Banten, Indonesia

* pbayu@yahoo.com

## Abstract

### Background

The primary challenge encountered by individuals diagnosed with endometriosis is the experience of pain. Emerging research indicates that oxidative stress is implicated in the initiation of pain associated with endometriosis. Vitamins C and E are known for their antioxidative properties. The primary objective of this study is to assess the efficacy of antioxidant supplementation, consisting of these vitamins, in the management of pain associated with endometriosis.

### Methods

A comprehensive search was conducted on the ClinicalTrials.gov, Scopus, Europe PMC, and Medline databases up until August 23rd, 2023, utilizing a combination of relevant keywords. This review incorporates literature that examines the relationship between antioxidant supplementation and pain in endometriosis. We employed fixed-effect models to analyze the risk ratio (RR) and present the outcomes together with their corresponding 95% confidence intervals (CI).

### Results

A total of five RCTs were incorporated. The results of our meta-analysis indicated that antioxidant supplementation with vitamin C and E combination was associated with higher proportion of endometriosis patients reporting reduced chronic pelvic pain (RR 7.30; 95%CI: 3.27–16.31, $p<0.00001$, $I^2 = 0\%$), alleviations of dysmenorrhea (RR 1.96; 95%CI: 1.25–3.07, $p = 0.003$, $I^2 = 39\%$), and dyspareunia (RR 5.08; 95%CI: 2.10–12.26, $p = 0.0003$, $I^2 = 0\%$) than patients only receiving placebo.

### Conclusions

This study suggests the potential ability of vitamin C and E in alleviating pain symptoms experienced by individuals with endometriosis.

**Data Availability Statement:** All relevant data are within paper and its Supporting Information files.

**Funding:** The author(s) received no specific funding for this work.

**Competing interests:** The authors have declared that no competing interests exist.

## 1. Introduction

On a global scale, it has been observed that approximately 10 to 15 percent of women within the reproductive age bracket have the medical condition known as endometriosis [1]. Endometriosis is a pathological condition characterized by the ectopic growth of endometrial tissue beyond the confines of the uterine wall [2]. In this particular pathological state, the endometrium exhibits the ability to proliferate within several anatomical sites, including the ovaries (ovaries), peritoneal lining of the stomach, intestines, vagina, or urinary system [2]. The endometrium refers to the specialized tissue that forms the inner lining of the uterus [2, 3]. Prior to menstruation, the endometrium undergoes a process of thickening, thereby creating a receptive environment for the implantation of a fertilized ovum [2, 3]. In the absence of fertilization, the endometrium undergoes shedding, resulting in the expulsion of menstrual blood from the body [2, 3]. In endometriosis, endometrial tissue that grows outside the uterus also thickens but lacks the ability to be shed and expelled from the body [2, 3]. This condition induces irritation or inflammation of the tissue surrounding the endometrium, resulting in patient complaints manifested as pain [2, 3].

The primary issue experienced by those with endometriosis is frequently pain [1, 4]. Pain may be experienced in the context of menstruation (dysmenorrhea), sexual intercourse (dyspareunia), or as a persistent pelvic discomfort occurring outside the menstrual cycle (chronic pelvic pain) [1, 4]. When managing the pain associated with endometriosis, healthcare professionals frequently prescribe analgesic medications such as acetaminophen or non-steroidal anti-inflammatory drugs (NSAIDs) [4, 5]. The administration of nonsteroidal anti-inflammatory medicines (NSAIDs) frequently leads to adverse effects on the gastrointestinal system, manifesting as abdominal pain, nausea, vomiting [6, 7]. In rarer instances, cardiovascular complications such as myocardial infarction (MI) and thromboembolic events may also occur [6, 7]. Nonsteroidal anti-inflammatory drugs (NSAIDs) can induce disruptions in renal hemodynamics, hence restricting their administration in individuals with impaired renal function [6, 7]. The aforementioned issue frequently represents the primary limitation of nonsteroidal anti-inflammatory drugs (NSAIDs) in the management of pain [6, 7].

Emerging research has indicated that the occurrence of oxidative stress is implicated in the development and progression of endometriosis [8]. Reactive oxygen species (ROS), which exhibit elevated levels in the presence of oxidative stress, are inflammatory agents that have the potential to induce cellular damage [8, 9]. Elevated levels of reactive oxygen species (ROS) have been linked to the synthesis of pro-inflammatory cytokines and prostaglandins by macrophages, as well as the stimulation of C-fibers through neurogenic inflammation [8, 9]. The involvement of peroxidized lipids and lipoproteins in endometriosis has also been well-documented [8, 9]. These mechanisms collectively contribute to the initiation of pain in individuals diagnosed with endometriosis [8, 9]. Antioxidants are implicated in the mitigation of reactive oxygen species (ROS) in order to potentially alleviate discomfort associated with endometriosis [8, 9]. A number of vitamins and minerals, including as vitamin A, vitamin C, vitamin E, zinc, copper, and selenium, have been identified as possessing antioxidant properties [10]. Among all of these, vitamin C and vitamin E are two distinct types of vitamins that possess antioxidant properties and exhibit minimal adverse effects, rendering them suitable for daily long-term consumption [10]. The co-administration of vitamin C and vitamin E is based on the concept of "vitamin E recycling," in which the antioxidant activity of oxidized vitamin E is effectively replenished by other antioxidants, such as vitamin C [10]. Moreover, the combination of vitamin C and vitamin E may increase the oxidation resistance of total serum lipids more efficiently than supplementation of vitamin E or vitamin C alone [10]. The objective of this study is to assess the efficacy of antioxidant supplementation with a mix of vitamin C and

vitamin E in the treatment of pain associated with endometriosis. We hypothesized that vitamin C and E supplementation can significantly reduce pain symptoms better than placebo in patients with endometriosis.

## 2. Materials and methods

### 2.1. Eligibility criteria

The present study has been carried out following the protocols and recommendations specified in the PRISMA statement [11]. The protocol of this review has been registered in PROSPERO (CRD42023459152). We incorporated all randomized controlled trials (RCTs) that examined the effectiveness of antioxidant supplementation, specifically a combination of vitamin C and vitamin E, in comparison to a placebo, for the purpose of alleviating pain symptoms in women diagnosed with endometriosis. We excluded studies from our current systematic review and meta-analysis if they: (1) did not clearly specify that the participants had been diagnosed with endometriosis; (2) were cell-based or animal studies; (3) did not include a placebo as the comparison group; (4) were not primary investigations; and (5) had not undergone the process of publication.

### 2.2. Search strategy and study selection

A comprehensive review of the literature was performed, focusing specifically on papers written in the English language. The search encompassed a time frame up until August 23$^{rd}$, 2023, and was undertaken across four prominent worldwide databases: Medline, Scopus, Europe PMC, and the ClinicalTrials.gov. The search terms utilized for the literature review were as follows: "(antioxidant OR anti-oxidant OR antioxidative OR vitamin C OR ascorbic acid OR L-ascorbic acid OR vitamin E OR alpha-tocopherol) AND (supplementation OR therapy OR administration) AND (endometriosis OR adenomyosis OR ectopic endometrial tissue OR ectopic endometrium)". More details regarding the search strategy for each database can be seen in S1 Table of S1 File. The initial search was conducted by two authors. The team also cross-checked the citations exported to the reference manager for consistency and completeness. To identify any additional relevant articles, citation tracking was performed by examining the references of the identified studies, tracking citations, and exploring related articles. Additionally, a follow-up search of gray literature sources was conducted. Titles and abstracts were screened independently by two authors, who excluded articles not relevant to the study. Full-text eligibility was conducted by the same two authors and the discrepancies were addressed in study judgments. S2 Table in S1 File contained the PRISMA 2020 checklist of the manuscript.

### 2.3. Data extraction

Two reviewers independently retrieved essential data extracted from the eligible articles including characteristics of participants (i.e., age, number of participants in each study arms) in addition to study characteristics (i.e., author last name, year of publication, country, study design, antioxidant content, and study duration). The data was tabulated into Microsoft Excel 2019.

The pain-related outcomes of this investigation were divided into chronic pelvic pain (everyday pain), pain during menstruation (dysmenorrhea), and pain during intercourse (dyspareunia). All of these outcomes were calculated as the proportion of patients who experienced decrease pain sensation during the last follow-up when compared to baseline.

### 2.4. Risk of bias assessment

The evaluation of potential bias in each study was conducted by two independent reviewers using standardized assessment tools. The Risk of Bias version 2 (RoB v2) was employed to assess the quality of each randomized trials [12]. This scale incorporates evaluations about the randomization of study participants, deviations from intended interventions, missing outcome data, measurement of the outcome, and selection of the reported results of the studies [12]. The authors' evaluations were categorized as "low risk," "high risk," or "some concerns" of bias [12].

### 2.5. Statistical analysis

The pain-related outcomes were calculated by using the Mantel-Haenszel formula to obtain the risk ratio (RR) along with the 95% confidence interval (95% CI). The I-squared ($I^2$; Inconsistency) statistic was employed to quantify the heterogeneity among research, where values exceeding 50% indicated a substantial or noteworthy level of heterogeneity [13]. The analysis will commence with the utilization of fixed-effect models. However, in the event that a substantial degree of heterogeneity is detected, a transition to random-effect models will be made. If the number of papers included in the meta-analysis exceeds 10, a funnel plot would be employed to evaluate the presence of publication bias. All analyses in this investigation were conducted using Review Manager 5.4, a software tool developed by the Cochrane Collaboration.

## 3. Results

### 3.1. Study selection and characteristics

In this review we searched 4 databases; Europe PMC (n = 673), Scopus (n = 53), PubMed Medline (n = 32), and ClinicalTrials.gov (n = 6). A total of 764 citations were retrieved for screening. We removed 742 citations as they were found to be duplicates and not eligible based on title/abstracts screening. Of 22 records screened for full-text eligibility, 17 studies were excluded based on the following reasons: seven articles were not using combination of vitamin C and vitamin E as the antioxidant component, six articles lacked data pertaining to the outcomes, and four articles were omitted as they were review articles. Ultimately, the remaining 5 randomized clinical trials (RCTs) [14–18] with a total of 338 endometriosis patients were included in the final analysis (Fig 1). One RCT had triple-blinding method, one RCT had double-blinding method, and the remaining three RCTs did not specify the blinding methods they used. All but one RCT used the combination of vitamin C 1000 mg/day and vitamin E 1200 IU/day as the antioxidant component. These antioxidant or placebo were given to the patients with endometriosis for a total of 8 weeks. A comprehensive overview of the characteristics of each study included in this analysis was provided in Table 1.

### 3.2. Quality of study assessment

Out of the five randomized controlled trials (RCTs) that were included in the analysis, only one demonstrated a "low" risk of bias across all areas of the assessment as determined by the RoB v2 criteria. The four remaining RCTs were classified as having a "some-concern" risk of bias. This classification was based on the absence of clear explanations on the randomization techniques employed and/or inadequate information regarding any deviations from the intended interventions that occurred owing to the specific circumstances of the experiment. Fig 2 provides a comprehensive overview of the risk of bias associated with each RCT.

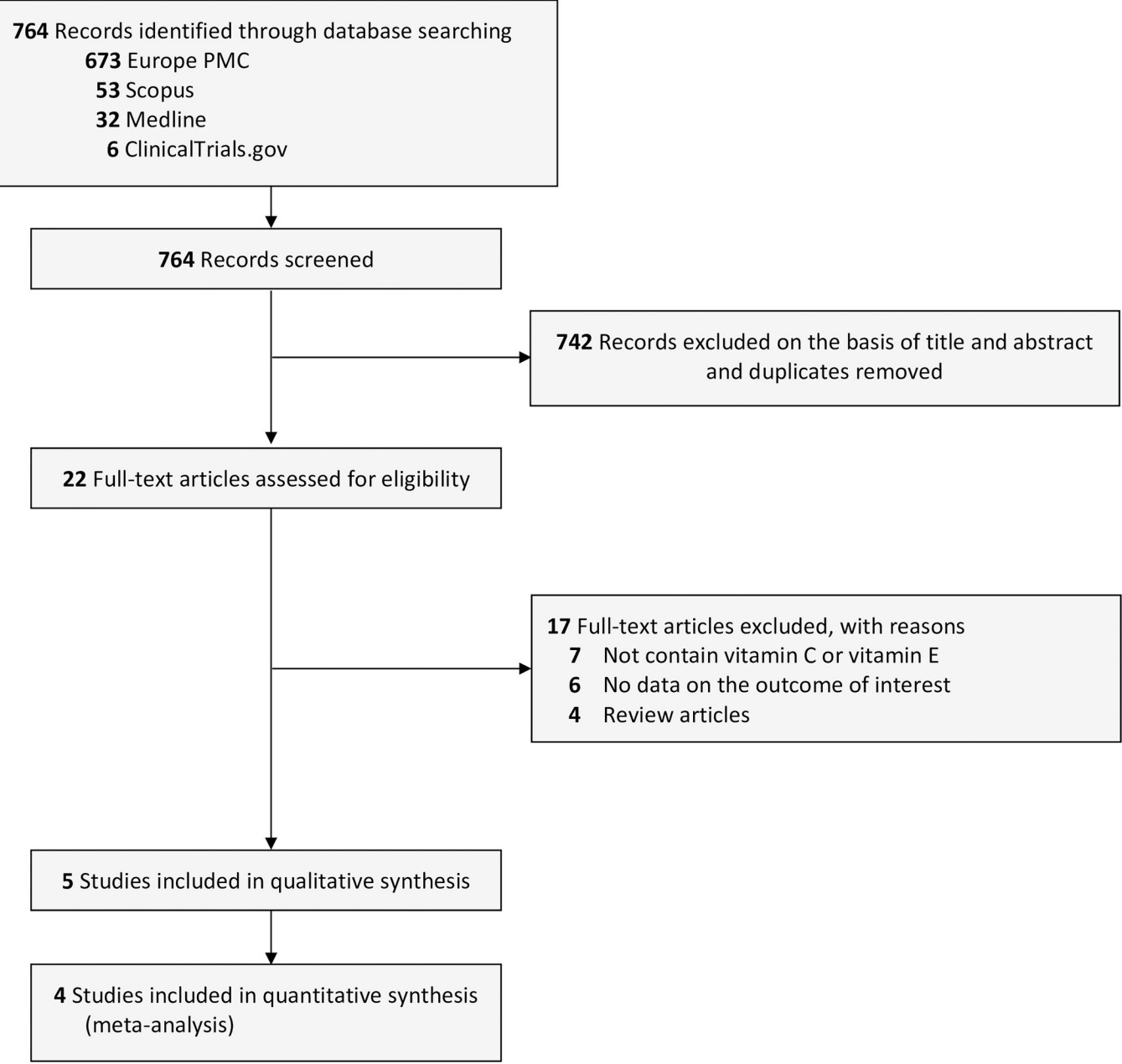

**Fig 1. Preferred Reporting Items for Systematic Reviews and Meta-Analysis (PRISMA) diagram of the detailed process of selection of studies for inclusion in the systematic review and meta-analysis.**

## 3.3. Pain-Related outcomes

**3.3.1. Chronic pelvic pain (Everyday pain).** The findings of the meta-analysis, which included four randomized controlled trials (RCTs), indicate that antioxidant supplementation with a combination of vitamin C and vitamin E is linked to a higher proportion of endometriosis patients reporting reduced chronic pelvic pain (everyday pain) sensation during follow-up, compared to those who were administered a placebo (RR 7.30; 95%CI: 3.27–16.31, $p < 0.00001$, $I^2 = 0\%$, fixed-effect model) (Fig 3A). One RCT by Amini L et al. [15] reported the chronic pelvic pain (everyday pain) in the numeric scale changes of visual analog score (VAS)

**Table 1. Characteristics of included studies that analyzed the relationship between antioxidant and pain symptoms in endometriosis patients.**

| Study | Country | Design | Antioxidant content | Study durations (weeks) | Study arms | Number of participants randomized | Age (years) |
|---|---|---|---|---|---|---|---|
| Al-Naggar MA et al. [14] 2022 | Egypt | Double-blind RCT | Vitamin C 1000 mg/day (in 2 tablets of 500 mg) + vitamin E 1200 IU/day (in 3 capsules of 400 IU) | 8 | Antioxidant | 30 | 32.5 ± 4.5 |
| | | | | | Placebo | 30 | 31.4 ± 5.2 |
| Amini L et al. [15] 2021 | Iran | Triple-blind RCT | Vitamin C 1000 mg/day (in 2 tablets of 500 mg) + vitamin E 800 IU/day (in 2 tablets of 400 IU) | 8 | Antioxidant | 30 | 35.7 ± 5.7 |
| | | | | | Placebo | 30 | 38.0 ± 6.4 |
| Kavtaradze N et al. [16] 2003 | USA | Prospective RCT | Vitamin C 1000 mg/day (in 2 tablets of 500 mg) + vitamin E 1200 IU/day (in 3 capsules of 400 IU) | 8 | Antioxidant | 46 | 19–41 |
| | | | | | Placebo | 13 | 19–41 |
| Santanam N et al. [17] 2013 | USA | Prospective RCT | Vitamin C 1000 mg/day (in 2 tablets of 500 mg) + vitamin E 1200 IU/day (in 3 capsules of 400 IU) | 8 | Antioxidant | 46 | 19–41 |
| | | | | | Placebo | 13 | 19–41 |
| Sehsah FIA et al. [18] 2022 | Egypt | Prospective RCT | Vitamin C 1000 mg/day (in 2 tablets of 500 mg) + vitamin E 1200 IU/day (in 3 capsules of 400 IU) | 6–8 | Antioxidant | 50 | 25.3 ± 3.7 |
| | | | | | Placebo | 50 | 26.1 ± 4.2 |

IU = international unit; RCT = randomized controlled trial; USA = United States of America

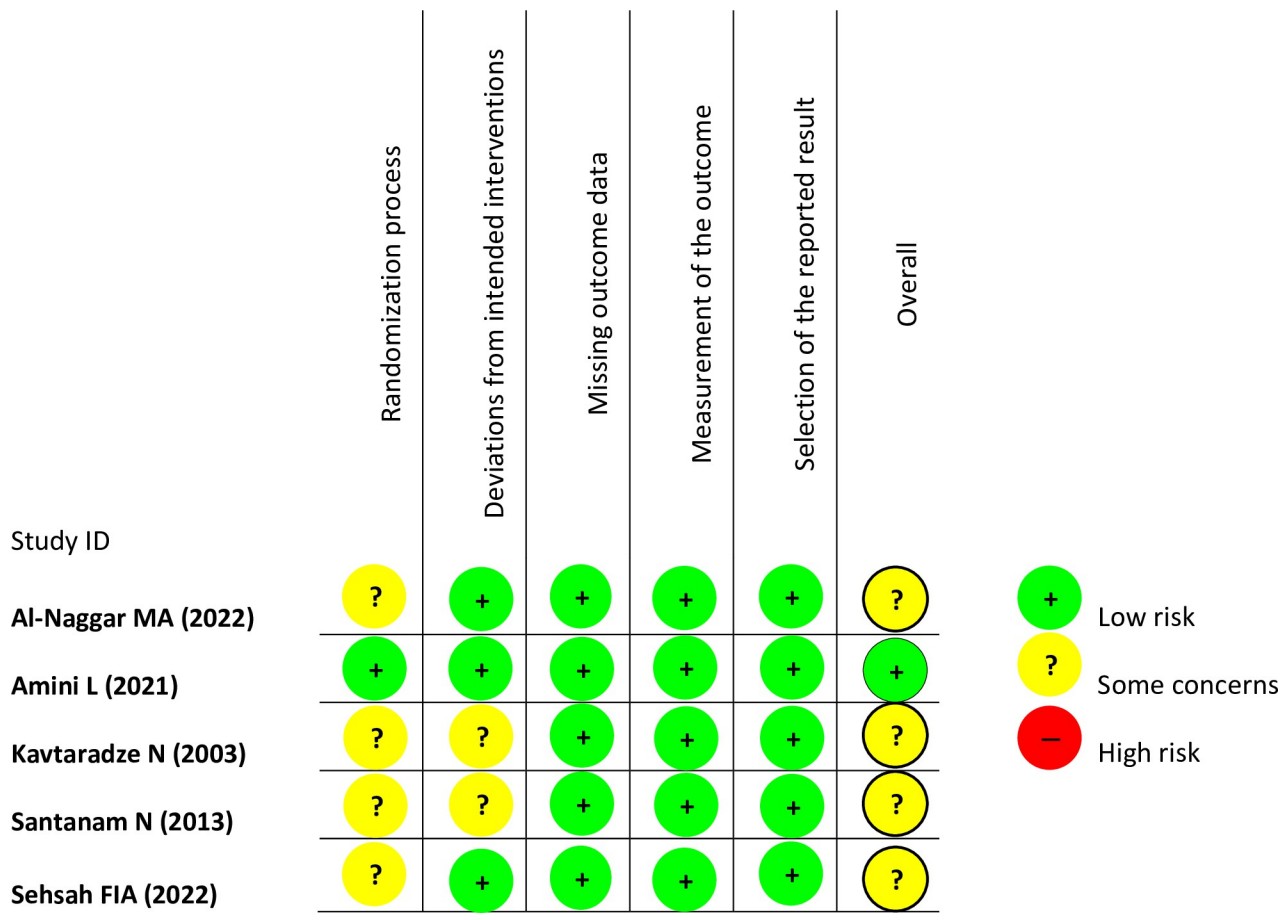

**Fig 2. Risk of Bias version 2 (RoB v2) quality assessment of clinical trial studies that shows some-concern risk of bias in four clinical trials and low-risk of bias in one clinical trial study.**

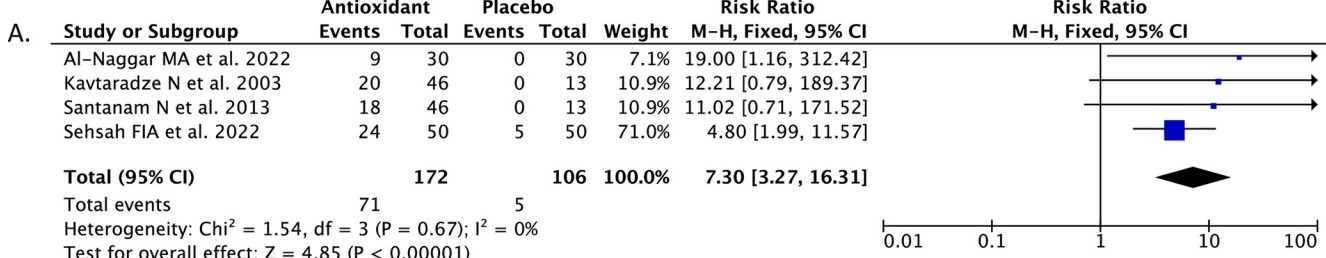

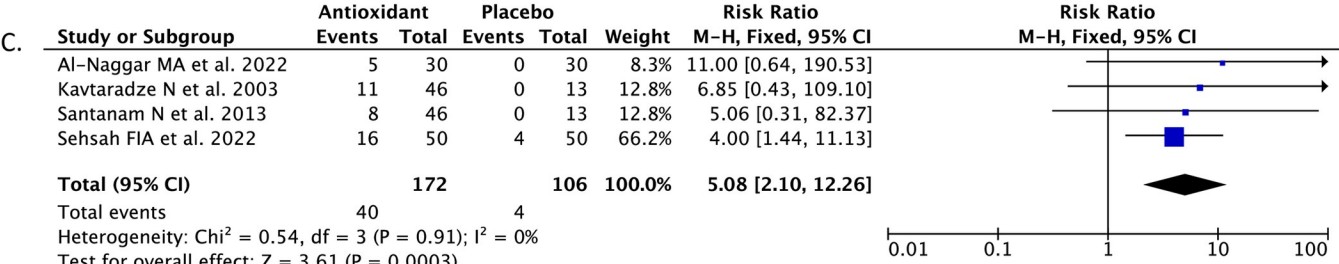

**Fig 3.** Forest plot that demonstrates antioxidant supplementation containing combination of vitamin C and vitamin E was associated with higher proportion of endometriosis patients who achieved reduction in the chronic pelvic pain (A), dysmenorrhea (B), and dyspareunia (C) symptoms.

from baseline to follow-up. In this RCT, the chronic pelvic pain in the antioxidant group was 66.26 ± 27.84 before treatment then significantly reduced to 12.43 ± 13.28 during the last follow-up at week 8 (p<0.001) [15]. In contrast, within the placebo group, the baseline chronic pelvic pain score was recorded as 16.96 ± 16.28. However, rather than exhibiting a decrease, this score increased to 18.63 ± 18.35 by the eighth week of follow-up (p = 0.571) [15]. These results indicate the favorability of antioxidants in reducing chronic pelvic pain caused by endometriosis.

**3.3.2. Dysmenorrhea.** The findings of the meta-analysis, which included four randomized controlled trials (RCTs), indicated a notable increase in the proportion of individuals who achieved a substantial alleviation of dysmenorrhea (menstrual discomfort) among those who were administered antioxidants comprising a mix of vitamin C and vitamin E, in comparison to those who got a placebo (RR 1.96; 95%CI: 1.25–3.07, $p = 0.003$, $I^2 = 39\%$, fixed-effect model) (Fig 3B). A RCT by Amini L et al. [15] reported the dysmenorrhea outcome in the numeric scale changes of visual analog score (VAS) from baseline to follow-up. In this RCT, the menstrual pain (dysmenorrhea) in the antioxidant group was 50.53 ± 32.12 before treatment then significantly reduced to 17.56 ± 16.65 during the last follow-up at week 8 (p<0.001). In contrast, within the placebo group, the baseline dysmenorrhea VAS score was recorded as 51 ± 34.21 and then significantly reduced to 31.56 ± 26.39 by the eighth week of follow-up

(p<0.001). However, the reductions in these VAS score of dysmenorrhea were greater in the antioxidant group than the placebo group (-32.37 vs. -19.44, p = 0.002). These results indicate the favorability of antioxidants in reducing menstrual pain (dysmenorrhea) caused by endometriosis.

**3.3.3. Dyspareunia.** The findings of the meta-analysis, which included four randomized controlled trials (RCTs), indicated a notable increase in the proportion of individuals who achieved a substantial alleviation of dyspareunia (discomfort during sexual intercourse) among those who were administered antioxidants comprising a mix of vitamin C and vitamin E, in comparison to those who got a placebo (RR 5.08; 95%CI: 2.10–12.26, $p$ = 0.0003, $I^2$ = 0%, fixed-effect model) (Fig 3C). In one included RCT that was conducted by Amini L et al. [15], the dyspareunia outcome was reported in the numeric scale changes of visual analog score (VAS) from baseline to follow-up. In this RCT, the sexual intercourse pain (dyspareunia) in the antioxidant group was 66.26 ± 28.27 before treatment then significantly reduced to 15.43 ± 18.47 during the last follow-up at week 8 (p<0.001). In contrast, within the placebo group, the baseline dyspareunia VAS score was recorded as 20.73 ± 21.77 and then slightly reduced to 18.1 ± 19.93 by the eighth week of follow-up (p<0.001). However, the reductions in these VAS score of dyspareunia were greater in the antioxidant group than the placebo group (-50.88 vs -2.63, p = 0.006). These results indicate the favorability of antioxidants in reducing menstrual pain (dysmenorrhea) caused by endometriosis.

## 3.4. Publication bias

Funnel plot analysis was employed to assess publication bias. The present investigation did not perform the assessment of publication bias due to the limited number of research included (less than 10 studies) in each outcomes of interest. Consequently, the evaluation of publication bias lacks the same level of robustness as when there are more than 10 studies available for analysis [19, 20].

## 4. Discussion

The findings of our meta-analysis have confirmed our initial hypothesis where antioxidant supplementation containing vitamin C and vitamin E can reduce the pain symptoms, ranging from chronic pelvic pain, dysmenorrhea, and dyspareunia better than placebo in patients with endometriosis. As aforementioned pathophysiological process of pain development in endometriosis where oxidative stress serves as the culprit, vitamin C and vitamin E can halt this process by scavenging excess reactive oxygen species (ROS) because of their antioxidant properties [9, 10].

Both vitamins are able to reduce the inflammatory process by inhibiting the production of several proinflammatory cytokines, such as tumor necrosis factor-alpha (TNF-α), interleukin-1 (IL-1), IL-6, and monocyte-chemotactic protein-1 which may be responsible for responsible for the release of pain-inducing molecules [9, 21, 22]. This inhibition of the inflammatory process is also obtained through the activity of vitamin C and vitamin E which reduce the activity of the cyclooxygenase (COX) enzyme so that it will reduce the process of converting arachidonic acid into prostaglandin E2 (PGE2) which is responsible for modulating pain [9, 23]. In addition to inhibiting the inflammatory process, these two vitamins also play a role in reducing oxidative stress due to excess iron or iron deficiency through the regulatory function of iron metabolism [9, 24, 25]. It is this mechanism that underlies the ability of the combination of vitamin C and vitamin E to reduce pain in patients with endometriosis. The background for combining vitamin C with vitamin E is the possibility of a synergistic effect of the two vitamins so that they can strengthen each other's ability to inhibit lipid peroxidation [26].

The findings of our study are consistent with previous research conducted by Sukan B et al. [27] and Zheng S et al. [28], which also demonstrate the efficacy of antioxidant vitamin supplementation in alleviating pain associated with endometriosis. However, both publications [27, 28] discussed antioxidant vitamins in general, while our current work focuses only on the antioxidant capacity of a combination of vitamin C and vitamin E. Moreover, there exist further distinctions between the present investigation and the prior study conducted by Sukan B et al. [27]. The preceding investigation conducted by Sukan B et al. [27] is a systematic review without meta-analysis. Systematic review studies would certainly be better if equipped with a meta-analysis because then we can get new data in the form of numbers that can estimate how much influence the intervention has when compared to controls/placebo. Previous study by Sukan B et al. [27] also does not specifically discuss the ability of vitamin C and vitamin E in reducing pain, but also discusses other substances that have antioxidant capabilities such as caffeic acid, resveratrol, garlic tablets, and a combination of N-acetyl cysteine, alpha lipoic acid and bromelain. Combining these different substances into the meta-analysis will certainly cause significant heterogeneity, so for our current study we only focus on antioxidant supplementation containing a combination of vitamins C and vitamin E. Of the eight studies included in the previous investigation by Sukan B et al. [27] there are only 3 studies that discuss vitamin C and vitamin E. The remaining five studies discuss other antioxidant substances as previously mentioned [27]. Meanwhile, our study included a total of 5 RCTs into the analysis, all of which discussed the ability of vitamin C and vitamin E for endometriosis.

There are various limitations inherent in our investigation. Initially, it is noteworthy that all randomized controlled trials (RCTs) incorporated in the study exhibit a comparatively modest sample size, amounting to less than 100 participants. Furthermore, the studies included in the study did not provide sufficient data on changes in pain scores on the Visual Analog Scale (VAS) from baseline to follow-up, hence precluding their inclusion in the meta-analysis. Information regarding the type of endometriosis (e.g. superficial/deep/ovarian) and any co-existing adenomyosis from the included studies is also lacking, therefore cannot be analyzed further. Furthermore, it is important to note that pain is a multifaceted symptom that is influenced by various factors such as the specific location of the disease (e.g., parametrial, sacral plexus), prior surgical procedures, concurrent medical conditions (e.g., interstitial cystitis), as well as peripheral and central sensitization [29]. Therefore, obtaining this information would greatly enhance the quality of the evidence presented. Regrettably, the included research do not contain this information. In addition, it is worth noting that the randomized controlled trials (RCTs) included in the study had a limited follow-up period of about two months. Consequently, the investigation failed to capture any potential long-term impacts of vitamin C and vitamin E supplementation, specifically in relation to both effectiveness and safety. Ultimately, it is worth noting that a majority of the randomized controlled trials (RCTs) examined in this study, specifically four out of five, exhibit a level of bias that can be categorized as "some-concern." This bias mostly stems from inadequate allocation concealment during the randomization process. Consequently, it is imperative to exercise caution when interpreting the findings of our investigation.

The findings of our systematic review and meta-analysis indicate a potential ability of antioxidant supplementation containing combination of vitamin C and vitamin E in mitigating the pain symptoms that may range from chronic pelvic pain, dysmenorrhea, to dyspareunia in individuals diagnosed with endometriosis. However, further research is required to validate the results of our study by the implementation of meticulously prepared randomized controlled trials (RCTs). These studies should ideally have bigger sample sizes and longer durations to ensure the robustness and reliability of the findings.

## Supporting information

**S1 File.**
(DOCX)

## Author Contributions

**Conceptualization:** Patrick Bayu, Jacobus Jeno Wibisono.

**Data curation:** Patrick Bayu, Jacobus Jeno Wibisono.

**Formal analysis:** Patrick Bayu, Jacobus Jeno Wibisono.

**Funding acquisition:** Patrick Bayu, Jacobus Jeno Wibisono.

**Investigation:** Patrick Bayu, Jacobus Jeno Wibisono.

**Methodology:** Patrick Bayu, Jacobus Jeno Wibisono.

**Project administration:** Patrick Bayu, Jacobus Jeno Wibisono.

**Resources:** Patrick Bayu, Jacobus Jeno Wibisono.

**Software:** Patrick Bayu, Jacobus Jeno Wibisono.

**Supervision:** Patrick Bayu, Jacobus Jeno Wibisono.

**Validation:** Patrick Bayu, Jacobus Jeno Wibisono.

**Visualization:** Patrick Bayu, Jacobus Jeno Wibisono.

**Writing – original draft:** Patrick Bayu, Jacobus Jeno Wibisono.

**Writing – review & editing:** Patrick Bayu, Jacobus Jeno Wibisono.

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
