## [Decision Letter · Decision Letter 0]

26 Dec 2023

PONE-D-23-32106Vitamin C and E antioxidant supplementation may significantly reduce pain symptoms in endometriosis: A systematic review and meta-analysis of randomized controlled trialsPLOS ONE

Dear Dr. Bayu,

Thank you for submitting your manuscript to PLOS ONE. After careful consideration, we feel that it has merit but does not fully meet PLOS ONE’s publication criteria as it currently stands. Therefore, we invite you to submit a revised version of the manuscript that addresses the points raised during the review process.

We look forward to receiving your revised manuscript.

Kind regards,

Diego Raimondo

Academic Editor

PLOS ONE

Reviewers' comments:

Reviewer's Responses to Questions

**Comments to the Author**

1. Is the manuscript technically sound, and do the data support the conclusions?

Reviewer #1: Yes

Reviewer #2: Yes

2. Has the statistical analysis been performed appropriately and rigorously? 

Reviewer #1: Yes

Reviewer #2: Yes

3. Have the authors made all data underlying the findings in their manuscript fully available?

Reviewer #1: Yes

Reviewer #2: Yes

4. Is the manuscript presented in an intelligible fashion and written in standard English?

Reviewer #1: Yes

Reviewer #2: Yes

5. Review Comments to the Author

Reviewer #1: I would like to extend my congratulations to all of you for the completion of your intriguing work. The manuscript demonstrates significant potential; however, I have a few concerns and suggestions that I believe could enhance the quality of your work.

1. Justification for the Choice of Antioxidant Vitamins: It is imperative that the authors provide a clear justification for their choice of combining vitamin C and E in their SRMA. While the manuscript highlights the efficacy of these specific vitamins, it is essential to address the presence of other antioxidant vitamins, such as vitamin A, zink and etc. A comprehensive rationale for this selection would strengthen the research.

2. I would like to bring to your attention that this is not the first SRMA regarding the use of antioxidant vitamins in reducing endometriosis-related pain. I recommend that the authors refer to PMC10464024.

3. Enhancement of the Discussion Section: The discussion section appears to be an area where further expansion is needed to align with the high standards of this journal. A more comprehensive and nuanced discussion would not only contribute to the scholarly value of the manuscript but also make it more engaging and thought-provoking for readers.

Reviewer #2: Thank you for giving me the opportunity to review this study.

Despite the good overall merit, I have some comments:

- include some details on the type of endometriosis (e.g. superficial/deep/ovarian) and any co-existing adenomyosis (e.g. doi 10.1002/jum.15237) of the selected studies, if available;

- pain is a very complex symptom (doi 10.1016/j.jmig.2022.10.007), it depends on several factors including localization of the disease (e.g. parametrial, sacral plexus), previous surgical interventions, coexisting conditions (e.g. interstitial cystitis), peripheral and central sensitization. Please include these issues in the Limitations section.

6. PLOS authors have the option to publish the peer review history of their article (what does this mean?). If published, this will include your full peer review and any attached files.

Reviewer #1: No

Reviewer #2: No

---

## [Author Response · Author response to Decision Letter 0]

6 Mar 2024

Dear Prof./Dr. Reviewers,

Thank you for your comments and suggestions regarding our manuscript/article. We already have corrected our article according to your comments.

Reviewer 1:

1) I would like to extend my congratulations to all of you for the completion of your intriguing work. The manuscript demonstrates significant potential; however, I have a few concerns and suggestions that I believe could enhance the quality of your work.

Answer: Thank you very much for your valuable comments and suggestions for our manuscript. We have revised our manuscript according to your suggestions.

2) Justification for the Choice of Antioxidant Vitamins: It is imperative that the authors provide a clear justification for their choice of combining vitamin C and E in their SRMA. While the manuscript highlights the efficacy of these specific vitamins, it is essential to address the presence of other antioxidant vitamins, such as vitamin A, zink and etc. A comprehensive rationale for this selection would strengthen the research.

Answer: Thank you very much for your suggestions. We have added a statement regarding other vitamins and minerals that possess antioxidant properties. We also have added statements regarding the rationale of combining vitamin C and vitamin E in the Introduction section. 

3) I would like to bring to your attention that this is not the first SRMA regarding the use of antioxidant vitamins in reducing endometriosis-related pain. I recommend that the authors refer to PMC10464024.

Answer: Thank you very much for your suggestions. We have revised that statement. We have mentioned about previous meta-analysis (PMC10464024). We also have provided brief comparison with previous studies on the almost similar topic.

4) Enhancement of the Discussion Section: The discussion section appears to be an area where further expansion is needed to align with the high standards of this journal. A more comprehensive and nuanced discussion would not only contribute to the scholarly value of the manuscript but also make it more engaging and thought-provoking for readers.

Answer: Thank you very much for your suggestions. Our Discussion is already divided according to the journal’s requirement and is already comprehensive. The first paragraph of our Discussion contained summary of the findings from our meta-analysis. The second paragraph of our Discussion contained explanations about the possible reasons behind the findings of our meta-analysis. The third paragraph of our Discussion contained comparison with the previous study. The fourth paragraph of our Discussion contained limitations from our study. The last paragraph of our Discussion contained conclusions from our study. Maybe if you have more evidence or statement to add for our Discussion section, we are very welcome.

 

Reviewer 2:

1) Thank you for giving me the opportunity to review this study.

Despite the good overall merit, I have some comments:

Answer: Thank you very much for your valuable comments and suggestions of our manuscript. We have revised our manuscript according to your suggestions.

2) include some details on the type of endometriosis (e.g. superficial/deep/ovarian) and any co-existing adenomyosis (e.g. doi 10.1002/jum.15237) of the selected studies, if available;

Answer: Thank you very much for your suggestions. However, that information is not available from any of the included studies. We have mentioned about this as one of limitations from our study. 

3) pain is a very complex symptom (doi 10.1016/j.jmig.2022.10.007), it depends on several factors including localization of the disease (e.g. parametrial, sacral plexus), previous surgical interventions, coexisting conditions (e.g. interstitial cystitis), peripheral and central sensitization. Please include these issues in the Limitations section.

Answer: Thank you very much for your suggestions. We have mentioned about this and cited the study that you have suggested as the limitation of our study.

If you still have any more concerns/comments/suggestions, please let us know. Thank you very much.

Best regards,

The Authors

---

## [Editor Report · Decision Letter 1]

25 Mar 2024

Vitamin C and E antioxidant supplementation may significantly reduce pain symptoms in endometriosis: A systematic review and meta-analysis of randomized controlled trials

PONE-D-23-32106R1

Dear Dr. Bayu,

We’re pleased to inform you that your manuscript has been judged scientifically suitable for publication and will be formally accepted for publication once it meets all outstanding technical requirements.

Kind regards,

Diego Raimondo

Academic Editor

PLOS ONE
---

## [Editor Report · Acceptance letter]

19 May 2024

PONE-D-23-32106R1 

PLOS ONE

Dear Dr. Bayu, 

I'm pleased to inform you that your manuscript has been deemed suitable for publication in PLOS ONE. Congratulations! Your manuscript is now being handed over to our production team.

Kind regards, 

on behalf of

Dr. Diego Raimondo 

Academic Editor

PLOS ONE